# Evaluating the feasibility, adoption, cost-effectiveness, and sustainability of telemedicine interventions in managing COVID-19 within low-and-middle-income countries (LMICs): A systematic review

**Nonye M. Okafor**[1], **Imani Thompson**[1], **Vandana Venkat**[1], **Courtney Robinson**[1], **Aishwarya Rao**[1,2], **Sumedha Kulkarni**[2,3], **Leah Frerichs**[2], **Khady Ndiaye**[2], **Deborah Adenikinju**[2], **Chukwuemeka Iloegbu**[2], **John Pateña**[2], **Hope Lappen**[3], **Dorice Vieira**[2,3], **Joyce Gyamfi**[2], **Emmanuel Peprah**[2*]

1 NYU School of Global Public Health, New York, New York, United States of America, 2 Department of Global and Environmental Health, Implementing Sustainable Evidence-based Interventions through Engagement (ISEE) Lab, NYU School of Global Public Health, New York, New York, United States of America, 3 NYU Health Sciences Library, NYU Grossman School of Medicine, New York, New York, United States of America

* ep91@nyu.edu

## Abstract

COVID-19 has tragically taken the lives of more than 6.5 million people globally, significantly challenging healthcare systems and service delivery, especially in low-and-middle-income countries (LMICs). This systematic review aims to: (1) evaluate the feasibility of telemedicine interventions for COVID-19 management; (2) assess the adoption of telemedicine interventions during the COVID-19 pandemic; (3) examine the cost-effectiveness of telemedicine implementation efforts and (4) analyze the sustainability of telemedicine interventions for COVID-19 disease management within LMIC service settings. We reviewed studies from selected public health and health science databases, focusing on those conducted in countries classified as low and middle-income by the World Bank, using telemedicine for confirmed COVID-19 cases, and adhering to Proctor's framework for implementation outcomes. Of the 766 articles identified and 642 screened, only 3 met all inclusion criteria. These studies showed reduced reliance on antibiotics, prescription drugs, and emergency department referrals among telemedicine patients. Statistical parity was observed in the length of stay, diagnostic test ordering rates, and International Classification of Diseases (ICD)-10 diagnoses between telemedicine and in-person visits. Telemedicine interventions designed for post-COVID physical rehabilitation demonstrated safety, sustainability, and enhanced quality of life for patients without requiring specialized equipment, proving adaptable across contexts with appropriate technology. These interventions were also economically sustainable and cost-effective for healthcare systems as a whole. Proposed strategies to bridge implementation gaps include community-level assessments, strategic planning, multisectoral partnerships of local hospital administration and lawmakers, legal consultations, and healthcare informatics improvements. Increased

**Data availability statement:** All relevant data are within the manuscript and its Supporting information files.

**Funding:** The author(s) received no specific funding for this work.

**Competing interests:** The authors have declared that no competing interests exist.

investment in telemedicine research focusing on infectious disease management is crucial for the continued development and refinement of effective strategies tailored to resource-constrained regions.

## Author summary

Our systematic review examines the feasibility, adoption, cost-effectiveness, and sustainability of telemedicine for managing COVID-19 in LMICs. Telemedicine has become an essential tool for expanding health management options while addressing healthcare delivery disruptions, physician shortages, and socio-economic barriers. In LMICs, limited healthcare access and resource constraints pose significant challenges. Key issues in implementing telemedicine include internet connectivity and device usage difficulties. However, telemedicine offers benefits such as technology applicability, financial savings, and improved health outcomes. Among the three studies reviewed, telemedicine's effectiveness was comparable to face-to-face consultations, particularly in post-COVID rehabilitation, which proved cost-effective and enhanced patients' quality of life. Telemedicine reduces healthcare access barriers and is adaptable across contexts with appropriate technology, but it can create challenges for diagnostics and medication access in rural areas, requiring a mix of in-person and telemedicine approaches. Despite these challenges, the benefits of telemedicine outweigh the drawbacks. We advocate for improving technological capabilities, creating community access points, and developing secure telemedicine platforms while prioritizing privacy and effective information sharing. There are opportunities for hospital teams, alongside regional and national lawmakers, to collaborate on research that enhances telemedicine strategies for managing COVID-19, emerging infectious diseases, and other health conditions in LMICs and globally.

## Introduction

The novel Coronavirus Disease 2019, commonly referred to as COVID-19 or SARS-CoV-2, has had a devastating global impact. Over 6.5 million lives have been lost, and more than 600 million individuals have been infected worldwide. The scale of this pandemic places it among the most catastrophic in modern history, echoing the severity of the 1918 Spanish Influenza [1,2]. Driven by its highly pathogenic nature and rapid transmission rate, COVID-19 poses substantial risks to human health. Extensive scientific research has identified SARS-CoV-2 as the causative agent, leading to a severe respiratory syndrome. This infectious disease is characterized by its transmissibility range, the potential to induce chronic morbidity, adverse symptoms, and, tragically, fatalities—particularly among vulnerable populations and underserved communities, including those residing in low and middle-income countries (LMICs) [3–5].

The strategic employment of evidence-based interventions (EBI), particularly the rapid development and widespread distribution of vaccines, has played a pivotal role in reducing morbidity and mortality associated with the SARS-CoV-2 virus. The multifaceted approach to research and the creation of vaccines encompassed a meticulous process of clinical trials, regulatory approvals, manufacturing at scale, and the implementation of extensive vaccination campaigns across diverse populations worldwide. Additionally, alongside vaccination efforts, public health measures, such as widespread testing, contact tracing, promotion of hygiene practices, and the implementation of social distancing protocols, significantly contributed to mitigating the spread and severity of the virus. This collective and multifaceted approach

marks a landmark in the global effort to combat the SARS-CoV-2 pandemic, showcasing the power of scientific innovation, public health coordination, and international collaboration in tackling a global health crisis of unprecedented magnitude.

While the global reduction of morbidity and mortality from the viral infection is a notable achievement, the advent of COVID-19 presented unprecedented challenges to healthcare systems and the provision of health services across all sectors. Its impact reverberated at various levels, inducing substantial shifts in individual behaviors, societal norms, and institutional operations [6]. The most discernible effects were experienced within underserved communities, particularly in LMICs. These countries, characterized by a gross national income (GNI) per capita between $1,036 and $4,045 [7], faced substantial economic disadvantages in implementing costly interventions to combat diseases within their resource-constrained settings [8].

Recognizing the urgency to curb the spread of COVID-19 and ensure the continuity of safe healthcare services, many nations turned to telemedicine interventions [9,10]. The implementation of telemedicine, especially in low-resource settings, provides a medium to increase access to primary preventative services, decelerate the progression of conditions into chronic diseases, and increase opportunities for accurate evaluation and prognosis of existing conditions [8]. Nevertheless, within LMICs, persistent economic and technological barriers continue to impede the feasibility of these interventions, consequently hindering their widespread adoption in these settings. Addressing these obstacles is essential to ensure equitable access to quality healthcare and technology, particularly in regions most vulnerable to the adverse impacts of such global health crises [8].

Telemedicine offers a promising avenue to mitigate disease transmission and safeguard vulnerable populations' health while extending crucial care to patients in need [6]. Harnessing advanced telecommunication strategies offers a way to mitigate geographical barriers to healthcare access that patients may experience while also protecting public health in the prevention of communicable diseases. Telemedicine is a platform that can connect patients with the necessary tools for health assessment, diagnosis, interventions, and consultations, transcending geographical barriers and enhancing access to healthcare services [11]. Telemedicine research in the context of COVID-19 still carries a degree of uncertainty.

Despite the novelty of COVID-19, there has been limited research published on how disease management affected crucial healthcare services as it pertains to the feasibility, adoption, cost-effectiveness, and sustainability of telemedicine interventions [8,10,12]. This ambiguity is particularly accentuated in the domain of COVID-19 treatment within LMICs, where challenges related to economic constraints, technological infrastructure, and healthcare resource availability further complicate the effective implementation and widespread adoption of telemedicine solutions. Addressing these uncertainties and challenges in LMICs is pivotal to ensuring equitable access to quality healthcare and fortifying healthcare systems against future health crises.

Telemedicine is a solution that ideally addresses a myriad of challenges posed by factors such as COVID-19-related restrictions, disruptions in routine healthcare delivery, physician shortages, and socio-economic barriers, all of which have significantly impeded access to essential care [13]. In LMICs, where prevailing healthcare challenges include limited accessibility to healthcare services and resource constraints, significant hurdles exist in effectively managing COVID-19 [14]. Introducing telemedicine interventions in this context may present challenges, but it is not an insurmountable endeavor.

The primary aim of this systematic review is to address the implementation outcomes of feasibility, adoption, cost-effectiveness, and sustainability of telemedicine interventions in managing COVID-19 within LMICs. In addition, we will address the following service outcomes: equity, efficacy, safety, and patient-centricity to understand the effects of telemedicine

interventions on specific patient populations in LMICs. This review lays the foundation for a vital research inquiry: How do the feasibility, adoption, cost-effectiveness, and sustainability of telemedicine interventions for managing COVID-19 impact the accessibility of essential healthcare services in LMICs? The significance of this review lies in its dedicated exploration of the unique challenges presented by COVID-19 within LMICs and its mission to pinpoint existing gaps in healthcare delivery and access within these communities. There exists an urgent need for further research into telehealth interventions for disease management on a global scale, with a particular emphasis on prioritizing LMICs in this crucial endeavor. Understanding and addressing these challenges are key to ensuring equitable healthcare access and fortifying healthcare systems, particularly in regions most vulnerable to the repercussions of global health crises.

## Methods

### Search strategy

We conducted a preliminary literature review to assess the impact of COVID-19 on LMICs. After gathering background information on the topic, intervention, and population, we developed our research question and formulated a systematic search strategy aimed at identifying published literature that met predefined inclusion criteria.

We formulated our search strategy based on the 2022 World Bank's income group classifications, which include low, lower-middle, upper-middle-, and high-income categories. These classifications were aligned with the most recent (August 29, 2022) Gross National Income (GNI) statistics to ensure accuracy [7]. We applied concise search strings like "telemedicine AND COVID-19 AND feasibility AND LMICS" and meticulously specified keyword spellings, such as "COVID-19" as opposed to "covid 19" or "covid19." This method was consistently applied across six scholarly databases: PubMed, CINAHL, Web of Science, Embase, Global Health, and Cochrane Library. The full search strategy is provided in S1 Appendix. The original search was conducted in October 2022 and was reemployed in November 2023 to ensure that no additional articles published during the authoring period were missed. The systematic review was conducted in accordance with the reporting guidelines outlined by the Preferred Reporting Items for Systematic Reviews and Meta-Analyses (PRISMA) 2020 checklist (S1 Checklist).

### Study criteria and data extraction

We utilized Covidence as our screening and extraction tool for all selected articles. Our study inclusion criteria encompassed articles conducted in LMICs that addressed positive COVID-19 cases and explored telemedicine applications for patients with COVID-19 in addition to other conditions, specifically focusing on randomized controlled trials (RCTs).

Covidence, an online tool designed for systematic reviews, played a central role in the title and abstract screening, full-text assessment, and data extraction processes, as illustrated in Fig 1. We imported the research information system (RIS) file format containing references into Covidence, which centralized all the references. This enabled our research team to evaluate each article's relevance to the research question meticulously.

During this screening phase, we established a rigorous two-researcher consensus requirement for articles to advance to the full-text review stage. In cases where a disagreement arose between researchers, a third reviewer provided the final resolution on whether to include the article. Subsequently, we conducted a comprehensive analysis, synthesizing the systematic review. This analysis was structured around Proctor et al.'s (2011) framework for implementation research outcomes [15], which encompassed feasibility, adoption, cost, and sustainability

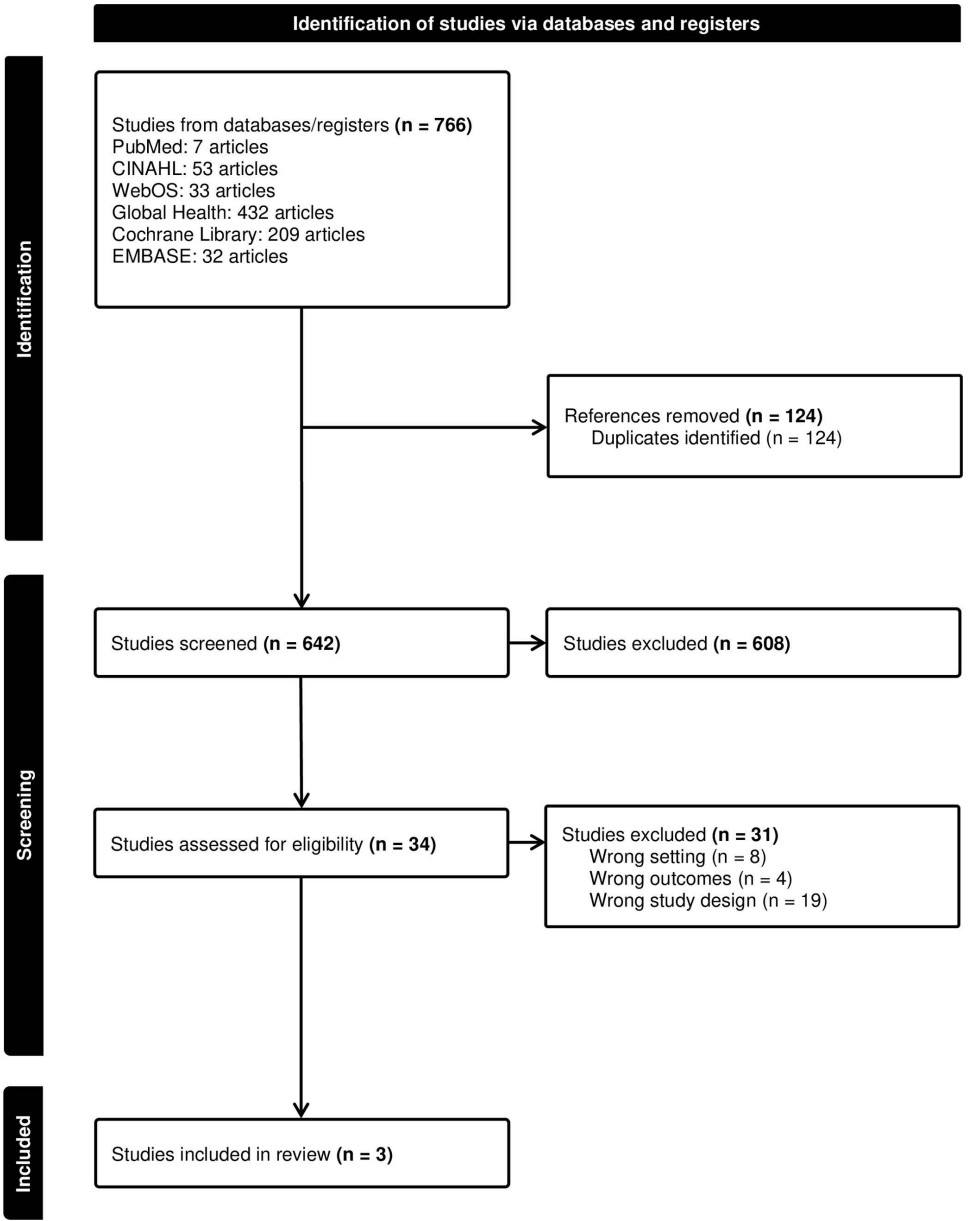

**Fig 1. PRISMA flow diagram generated in Covidence.**

(Table 1). These specific outcomes were selected as they provide insight into the barriers LMICs face in implementing telemedicine-based healthcare delivery services and pinpoint potential root causes, including available resources and current infrastructure. This framework was employed to assess the effectiveness of telemedicine interventions for COVID-19 disease management and treatment in LMICs.

## Risk of bias

We conducted a rigorous assessment of the risk of bias and the overall quality of the included articles, adhering to the guidelines outlined in the work by Hong et al. (2018) [17]. This

**Table 1. Implementation research and service outcomes definitions.**

| Outcomes | Definitions |
|---|---|
| Feasibility (IR) | "The extent to which the innovation can be successfully used or carried out within a given agency or setting" [15] |
| Adoption (IR) | "The intention, initial decision or action to try or employ the innovation (i.e., uptake)" [15] |
| Cost (IR) | "(Incremental or implementation cost) is defined as the cost impact of an implementation effort" [15] |
| Sustainability (IR) | "The extent to which a newly implemented innovation is maintained or institutionalized within a service setting's ongoing, stable operations" [15] |
| Efficacy (SO) | "Provision of services based on scientific knowledge to all who could benefit and refraining from providing services to those not likely to benefit (avoiding underuse and overuse)" [15,16] |
| Safety (SO) | "The avoidance of injuries to patients from the care that is intended to help them" [15,16] |
| Equity (SO) | "Provision of care that does not vary in quality because of personal characteristics such as gender, ethnicity, geographical location and socioeconomic status" [15,16] |
| Patient – centricity (SO) | "Provision of care that is respectful of and responsive to individual patient preferences, needs, and values, while also ensuring that patient values guide all clinical decisions" [15,16] |

Based on the taxonomy of implementation outcomes defined by Proctor et al. [15] and translated by the landmark study 'Crossing the Quality Chasm: A New Health System for the 21st Century' by the Institute of Medicine [16].

IR = Implementation research outcomes; SO = service outcomes.

Definition: Impact = Positive or negative, health improvements, any health indicator in disease management, treatment, and COVID-19 diagnosis.

comprehensive approach involved using the Mixed Methods Appraisal Tool (MMAT) to evaluate different methodologies across two domains: qualitative and quantitative research. The scores assigned were based on specific criteria, with each study having the potential to receive a maximum score of 24. Our assessment revealed that all three articles incorporated into the review attained a score of 21 out of 24, reflecting that the studies met the quality assessment criteria and limited the risk of bias (Table 2).

To ensure the objectivity and robustness of our assessment, two independent reviewers meticulously evaluated the selected studies and documented their findings within the MMAT tool. The colored cells represented three distinctive levels of methodological pattern and quality based on the studies' information and the criteria in each category of the tool. Green indicated high adherence to criteria with 2 points, yellow signified partial adherence or gaps with 1 point, and red reflected non-adherence with 0 points. Subsequently, a constructive dialogue took place among the researchers to address any discrepancies, ultimately culminating in the creation of a final quality assessment table (Table 2).

## Results

### Literature search

A total of 766 articles were imported into Covidence, with PubMed yielding 7 articles, CINAHL 53, Web of Science 33, Global Health 432, Cochrane Library 209, and Embase 32 (Fig 1). After removing 124 duplicate articles during this initial screening process, the remaining articles underwent a title and abstract screening. During this phase, researchers evaluated whether articles addressed the research question and aligned with the established inclusion and exclusion criteria. Throughout this meticulous screening process, 608 studies were deemed irrelevant, leading to the selection of 34 studies for full-text review. However, upon conducting the full-text review, 31 studies were subsequently excluded for various reasons, including incorrect study designs (n = 19), inappropriate settings (n = 8), and irrelevant

**Table 2. Mixed methods appraisal tool (MMAT).**

| Category | Criteria | Accorsi et al. | Pehlivan et al. | Ji et al. |
|---|---|---|---|---|
| Screening questions (For all types) | S1. Are there clear research questions? | Yes | Yes | Yes |
| | S2. Do the collected data address the research questions? | Yes | Yes | Yes |
| Qualitative assessments | 1.1. Is the qualitative approach appropriate to answer the research question | Yes | Yes | Yes |
| | 1.2. Are the qualitative data collection methods adequate to address the research question? | Yes | Yes | Yes |
| | 1.3. Are the findings adequately derived from the data? | Yes | Yes | Yes |
| | 1.4. Is the interpretation of results sufficiently substantiated by data? | Partially | Partially | Partially |
| | 1.5. Is there coherence between qualitative data sources, collection, analysis and interpretation? | Yes | Yes | Yes |
| Quantitative Assessments (Randomized Control Trials [RCTs]) | 2.1. Is randomization appropriately performed? | Yes | Yes | Yes |
| | 2.2. Are the groups comparable at baseline? | Yes | Yes | Yes |
| | 2.3. Are there complete outcome data? | Partially | Partially | Partially |
| | 2.4. Are outcome assessors blinded to the intervention provided? | Partially | Partially | Partially |
| | 2.5 Did the participants adhere to the assigned intervention? | Yes | Yes | Yes |
| | Total score | 21 | 21 | 21 |
| | Legend | Yes = 2 | No = 0 | Partially = 1 |

outcomes (n = 4). Ultimately, the review retained three articles conducted in middle-income countries: Brazil, China, and Turkey, respectively.

## Implementation outcomes

**Feasibility, adoption, cost-effectiveness, and sustainability.** Accorsi et al. assessed the adoption outcome (Table 1), examining the accuracy of the COVID-19 diagnostic approach and treatment delivery on patients in Brazil. Their study primarily scrutinized the comparison between telemedicine (67.4%) and face-to-face (72.1%) assessments of upper respiratory infections and symptom diagnosis based on groupings of the most prevalent disease classification, which revealed statistically similar results Kappa 0.386 [95% CI: 0.112-0.66]; p = 0.536 (Table 3) [18].

Moreover, with the adoption of telemedicine interventions, researchers observed a remarkably high level of agreement (Kappa = 0.715 [95%CI: 0.413–1]) in RT-PCR tests ordered by healthcare providers for detecting COVID-19 based on the symptoms described by patients. The rate among telemedicine patients (76.5%) was statistically similar (p>0.999) to those who received face-to-face evaluations (79.4%), detailed in Table 3 [18]. Conversely, the study revealed an observable difference in the prescription of antibiotics with telemedicine interventions (5.9%) resulting in fewer antibiotic prescriptions than face-to-face interventions (17.6%).

Pehlivan et al. evaluated telerehabilitation for COVID-19 treatment, focusing on the primary implementation outcomes of feasibility, sustainability, and cost in Turkey (Table 3) [19]. Their research uncovered that telerehabilitation emerged as an effective intervention, requiring no specialized equipment and demonstrating practicality compared to traditional rehabilitation methods. This intervention promoted sustainability by establishing and laying out its context and process in a clear manner for patient understanding, including gaining informed consent (IC) and operating under observation by a local ethics committee. Online-suitable tests were selected to evaluate participants, and the scoring of the tests was done by licensed professionals. Additional factors that contributed to overall sustainability included: ongoing

**Table 3. Implementation research outcomes for COVID-19 telemedicine interventions.**

| Authors | Study location (study period) | Study population | Intervention and aim | Implementation research outcomes | Service outcomes |
|---|---|---|---|---|---|
| Accorsi, TAD; Moreira, FT; Pedrotti, CHS; Amicis, K; Correia, RFV; Morbeck, RA; Medeiros, FF; Souza, JL; Cordioli, E | São Paulo, Brazil (September 2020 and November 2020) | Participants (n = 98) with suspected respiratory tract infection were enrolled >18 < 65 years old. Mean age: 36.3 ± 9.7 Gender distribution: 57.1% (n = 55) were female and 43.9% (n = 43) were male Study Group distribution: Telemedicine group (n = 48) and Non-telemedicine group (n = 50) | Telemedicine [TM-ED] (experimental) and non-telemedicine [ED only] (control group) Aim: Compare telemedicine diagnosis with face-to-face evaluation in patients with suspected respiratory tract infection who spontaneously sought evaluation at an emergency department. | **Adoption:** Diagnostic distribution by Telemedicine in TM-ED Group was 29 (67.4%) for upper airway infection and was statistically similar to the subsequent face-to-face assessment, 31 (72.1%), Moderate Kappa 0.386 [95% CI: 0.112–0.66]; p = 0.536. COVID-19 RT-PCR was ordered by TM in 26 (76.5%) versus 27 (79.4%) in face-to-face evaluation, High Kappa 0.715 [95%CI: 0.413–1]; p>0.999. Telemedicine trended towards less antibiotic prescription: 2 (5.9%) vs. 6 (17.6%), Moderate Kappa 0.452 [95%CI: 0.021 – 0.873]; p = 0.125. | **Efficacy:** In this study involving patients with acute respiratory symptoms who had sought evaluation at an ED, those who received TM evaluation were markedly more quickly evaluated (median of 5 minutes and 30 seconds of remote assistance). **Safety:** Another significant TM advantage in low-risk patients suspected of RTI is reduction of exposure to contagion at ED and the need for personal protective equipment. No clinical adverse events were reported associated with this study. **Equity:** There was no statistical difference between the TM-ED and face-to-face ED groups concerning diagnostic tests, ED length of stay, final ED grouped ICD-10 diagnosis, and prescription. The TM-ED Group had a similar patient profile and regular care practice to the ED Group. **Patient – centricity:** All patients provided written informed consent. Two patients withdrew consent. Four patients in the TM-ED Group were called for face-to-face evaluation before TM, and one had connection problems. These patients were analyzed on an intention-to-treat basis. |
| Pehlivan, E.; Palali, I.; Atan, S. G.; Turan, D.; Cinarka, H.; Cetinkaya, E. | Turkey (2020) | Participants (n = 34 patients) diagnosed with COVID-19, discharged within 4 weeks, and completed the program were evaluated >18 <75 years old. Mean age: 47 years Gender Distribution: 73.5% (n = 25) were males, 26.5% (n = 9) were females Study Group distribution: Telerehabilitation group (n = 17) and Non-telerehabilitation group (n = 17). | Telerehabilitation [TeleGr] (experimental) and non-telemedicine [CGr] (control group) Aim: Investigate the effectiveness of a telerehabilitation exercise program performed without requiring any special equipment on the physical condition of COVID-19 subjects from home. | **Feasibility:** Lower SGRQ scores indicate better health. In our study, there was an improvement (43.76% decrease) in SGRQ activity, impact, and total scores before (31.47) and after (17.70) in TeleGr. There was less improvement (26.28% decrease) observed in the CGr scores. As a result, the telerehabilitation exercise program has been shown to improve the quality of life of COVID-19 cases and is an alternative method that can be suitable for situations, including pandemics. **Sustainability:** The evaluations of the participants were carried out online due to the pandemic conditions by live videoconferencing. Therefore, evaluations were selected from tests that are suitable for online evaluation. Participants were asked to do their exercises 3 days a week for 6 weeks. Among 61 people who were invited, 40 participants completed the rehabilitation program (65% response rate). A total of 6 cases were dropped during follow up and final analysis, leaving 34 subjects to be examined. **Cost:** Telerehabilitation is a low-budget and specialized, tool-free intervention from home. It is noteworthy that pre- and post program evaluations were carried out at the hospital. | **Efficacy:** A significant improvement was observed in TelerGr in terms of mMRC (P = 0.035), TUG (P = 0.005), 5 sit-to-stand time which is one of the subtests of SPPB (P = 0.039) and SGRQ: activity (P = 0.003), impact (P = 0.005), and total scores (P = 0.002). There was only an improvement in the pain score of the CGr (P = 0.039). SGRQ activity (P = 0.035) and total scores (P = 0.042) further improved in TeleGr compared to CGr, according to the comparison of the delta values. **Safety:** All subjects were under follow-up by a pulmonologist and received their optimal medication. Considering the absence of any harmful effects of the program, it can be said that the telerehabilitation exercise program is safe and useful. **Equity:** The telerehabilitation exercise program performed without requiring any special equipment using live videoconferencing led to improved symptom scores, increased physical performance, and improved quality of life of post-COVID-19 patients. **Patient - centricity:** In all exercises, the patients were asked to place the camera in a place, where the patient could be easily seen by the physiotherapist. Subjects' relatives were requested to be in the room during the exercise sessions. Exercise was interrupted in case of excessive fatigue, palpitations, dyspnea, and the subjects' request to interrupt exercise. |

*(Continued)*

**Table 3.** (Continued)

| Authors | Study location (study period) | Study population | Intervention and aim | Implementation research outcomes | Service outcomes |
|---|---|---|---|---|---|
| Ji, W; Shi, W; Li, X; Shan, X; Zhou, J; Liu, F; Qi, F | China (2022) | Participants (n = 1500) were permanent community residents with absence of psychiatric history and cognitive impairment; participation was voluntary and required informed consent of the participant or guardian. Mean age: 44 years Gender distribution: 877 (58.5%) males, 623 (41.5% females) Study Group distribution: study was conducted with the test group (n = 1500) itself as a control (n = 1500); there was no sample loss. | Telemonitoring group (experimental); same group after 6 months (control group) Aim: To propose a community health intervention with remote monitoring and teleconsultation during COVID-19 for the prevention and control of COVID-19 at the rural level. | **Adoption:** Community health service centers were equipped with intelligent examination and monitoring equipment and established an internet management platform—dynamic monitoring and treatment guidance. Community residents used cell phones, computers, and other electronic devices to instantly transmit health data to the community health monitoring and management platform, forming dynamic electronic health records. Participants assessed the benefits of remote monitoring regarding medical services, information systems, financial costs, and psychological care via a comprehensive questionnaire split in two parts. **Feasibility:** The results showed that the remote monitoring intervention significantly (p<0.001) improved participants' healthy cognitive and behavioral efficacy in the grassroots community. The mean difference of the scores in favor of remote monitoring were the following: Cognitive awareness (3.54 ± 1.85), Perceived benefits (20.30 ± 3.41), Self-efficacy (5.90 ± 2.24), and Behavioral outcomes (7.74 ± 2.54). **Cost:** Limited data was provided, however, it was discussed that remote monitoring and consultation ensure the accessibility of primary health care and take advantage of online information technology; as well as the low costs to conduct online psychological counseling without threatening the health and safety of the people. | **Efficacy:** This study showed a significant (p<0.001) difference in the mean scores of participants' self-efficacy before (16.08 ± 4.13) and after (21.98 ± 1.89) the telemonitoring intervention in the community. After the intervention, residents were more willing to engage in COVID-19 prevention activities, such as taking the initiative to fill in health information, strengthening personal protection, and reducing going out. Through remote monitoring and control, remote consultation, and graded treatment, the quality of public health services has been significantly improved. The risk rate of COVID-19 exposure has been reduced to ensure the health quality of the community. **Safety:** Effective avoidance of cross-infection is an essential tool for primary outbreak prevention and control. The teleconsultation service of graded consultation can meet the needs of the home-isolated population for daily consultation. Telemedicine allowed for safe and efficient information acquisition methods and transmission channels. **Equity:** In this intervention, participants were asked to fill in their health information on their cell phones once a day. For the elderly living alone, left-behind children, and other people with difficulties using intelligent devices, community control staff of local community service stations provided assistance for filling in health information and household monitoring services for this group of people after receiving standardized training. Overall, telemedicine reduced medical barriers in rural areas, especially in COVID-19 monitoring. **Patient - centricity:** The remote monitoring intervention helps community residents achieve real-time monitoring and feedback of their health status through cell phone terminals, with quick access to information related to medical services along with accurate and comprehensive content. |

NR = not reported; TM = telemedicine; ED = emergency department; TeleGr = telerehabilitation group; CGr = control group; mMRC = modified Medical Research Council; SPPB = Short Physical Performance Battery; SGRQ = Saint George Respiratory Questionnaire (SGRQ); TUG = Timed "Up and Go" test.

communication between the practitioners and participants, regular evaluation, adherence to visit schedule, and follow-up, resulting in a strong study response rate of 65%. While the intervention is cost-friendly since it requires no special equipment, it is not cost-free; there is minimal financial burden associated with telerehabilitation benefits for both patients and hospital administrations, particularly in the context of follow-ups and ongoing care delivery [20].

The results of the feasibility assessment revealed significant overall improvements in the modified Medical Research Council (mMRC) (p = 0.035), Timed "Up and Go" test (TUG) (p = 0.005), Short Physical Performance Battery (SPPB) 5-times sit-up time (p = 0.039), and Saint George Respiratory Questionnaire (SGRQ) activity (p = 0003), impact (p = 0.005), and total scores (p = 0.002) within the telerehabilitation intervention group (TeleGr) compared to only one significant improvement in the control group (CGr) in their pain score (p = 0.039) at the end of the program (Table 3) [19]. Furthermore, an indicator of quality of life, the SGRQ total score in the TeleGr group demonstrated a more notable decrease (43.76%) in health-related impairments of subjects' quality of life compared to the CGr group (17.70%) before and after treatment.

Ji et al. assessed the adoption, feasibility, and cost considerations of remote monitoring and teleconsultations for COVID-19 prevention in rural communities in China (Table 3) [21]. Remote monitoring was referred to as the use of wearable or mobile technology to track patients' health status, allowing providers to monitor a patient's disease progress closely and from anywhere. Teleconsultations were referred to as a way of connecting patients to medical specialists to discuss their care and treatment or receive guidance virtually, in their current location or hospital. Both are subsets of telemedicine and were used in tandem throughout the study. To facilitate the uptake and adoption of the telemedicine interventions, community health centers were equipped with workstations, monitoring equipment, and an internet management platform, enabling remote examinations, treatment guidance, and dynamic monitoring. A noteworthy outcome was the enhanced efficiency of consultations at the community hospital due to remote registration, which helped reduce cross-infection risks within the rural population. Patients completed a valid and reliable questionnaire to collect their demographic data and sentiments on the effectiveness of remote monitoring at the start and end of the program. Furthermore, after 6 months, the feasibility assessment revealed that the remote monitoring intervention significantly (p<0.001) improved patients' cognitive awareness (positive mean difference: 3.54 ± 1.95), perceived benefits (positive mean difference: 20.30 ± 3.41), self-efficacy (positive mean difference: 5.90 ± 2.24), and behavioral outcomes (positive mean difference: 7.74 + 2.54) (Table 3) [21].

Telemedicine played a pivotal role in encouraging self-quarantine while ensuring access to appropriate care across various settings [22–24]. This approach substantially reduced hospital foot traffic, eliminated the need for in-person patient registration, and effectively controlled and prevented COVID-19 outbreaks (Table 3) [21]. Additionally, the study demonstrated that teleconsultations decreased the labor demands on hospital personnel and fostered diverse interactions and connectivity between patients and providers across great distances. This approach positively impacted patient health status while also saving time and resources (Table 3).

Although cost assessment data were limited, the study highlighted that remote monitoring and teleconsultations represented a cost-effective method, capitalizing on readily available technology and promoting the physical and psychological well-being of rural populations. The results underscored positive changes in the cognitive and behavioral outcomes of the participants in the study, which created downstream effects, including increased community health awareness, establishment of dynamic electronic health records, enhanced telemedicine services, health education, and promotion through the study. During COVID-19, in rural communities, telemedicine interventions improved community health and reduced medical barriers overall [21].

## Service outcomes

**Efficacy, safety, equity, and patient-centeredness.** These secondary service outcomes encompassed health service indicators, including reported efficacy, safety, equity, and patient-centricity for each trial. Definitions for these service outcomes are provided in Table 1. Among all three articles, patient-centric efforts were most frequently discussed compared to other service outcomes, followed by equity, and safety.

Pehlivan et al.'s study, for instance, emphasized patient-centricity through timely and routine follow-ups by a pulmonologist, along with optimal medication distribution, patient education programs, and a flexible approach accommodating patient interruptions due to fatigue or other health events during the telerehabilitation intervention (see Table 3) [19]. Research equity was maintained by providing readily accessible programming for participants, and safety was evaluated through the ongoing monitoring of program effects.

Ji and colleagues articulated their patient-centered approach by engaging in the discretionary collection of health information from rural residents participating in remote monitoring and telemedicine consultations. This approach involved remote and off-site face-to-face health monitoring to enhance access to specialized medical services during COVID-19, coupled with upgrading technical supplies, treatment offices, and medical equipment in community health centers (Table 3) [21]. In ensuring that patients of all ages were equipped with assistance for utilizing devices intended to monitor symptoms based on need, Ji et al. increased health equity and reduced medical barriers commonly experienced in rural areas. They also found that real-time self-monitoring of symptoms completed by patients yielded increased visibility and identification of those believed to be at risk for having been infected by COVID-19. This, in turn, influenced the overall safety of participants by reducing the risk of cross-infection and outbreak.

In Accorsi et al.'s study, patient-centric efforts were characterized by the consolidated implementation of face-to-face procedures and data collection during emergency department stays (18.7%) to minimize travel inconveniences. Additionally, the routine care options received did not vary in quality across treatment groups for diagnostic tests (81.6% total), average ED length of stay (89.5 ± 88.5 minutes), ICD-10 diagnosis for upper airway infection (70.4% total), and medical prescriptions (23.5% total); both groups had similar patient profiles (see Table 3) [18]. A reduction in exposure and spread of COVID-19 provided a protective advantage, increasing the safety of participants within the emergency department.

## Facilitators and barriers to implementation

Our investigation delved into the methods and tools employed to facilitate the implementation of telemedicine interventions while mitigating existing barriers. This exploration unveiled that telemedicine interventions have evolved into a multifaceted approach catalyzed by the contemporary surge in accelerated telecommunications services and delivery [25]. This transition has ushered in convenience and expanded health management options for both patients and medical practitioners, aligning with prior research findings [18].

We identified three pivotal facilitator categories within this context: technology, financial considerations, and personal attitudes. Heightened technology usage, exemplified by increased mobile and internet utilization, has enabled the seamless implementation of telemedicine. For instance, the study by Accorsi et al. demonstrated the significant role of telemedicine in detecting and managing patients' conditions, curbing the transmission of COVID-19, and adhering to international health guidelines for COVID-19 management [18].

Considering financial factors, telemedicine exhibited cost-effectiveness and minimal equipment requirements, primarily necessitating only a device, such as a smartphone, for

accessibility. Consequently, the utilization of this intervention emerged as an affordable, cost-effective means of health management adoption and feasibility within LMICs (Table 3), which translated into improved outcomes in symptom scores, physical performance, and quality of life for patients, as demonstrated by Pehlivan et al. [19].

Furthermore, Ji et al.'s study highlighted that personal and community sentiments regarding COVID-19 prevention bolstered the adoption, cost-effectiveness, and feasibility of remote monitoring through telemedicine (Table 3). Through telehealth assessments, it became evident that telemedicine effectively curtailed the spread of COVID-19 by limiting patients' exposure to hospital/clinic environments, thereby minimizing the risk of cross-infection. Additionally, telemedicine fostered positive community behaviors towards preventive health measures, contributing to public health management and awareness. This approach also promoted cost-effective access to online health services and resources while preventing the spread of COVID-19 infections [21]. Amidst these common facilitator themes, there were notable barriers to telemedicine implementation in LMICs, including reports of internet connectivity issues and challenges in using technological devices.

## Quality of evidence

The quality of evidence of telemedicine interventions in LMIC settings was based on the results reported across our referenced studies. To aid in this evaluation, the implementation outcomes were measured through standardized methodology, data sources, and the observed intersections of our outcome measurements across these studies.

Study implementation outcomes were standardized relative to the study aims using the Proctor et al. framework. For example, within the implementation outcome of adopting telemedicine interventions, Accorsi et al. observed differences in tests and prescriptions ordered by providers based on patient symptoms when measuring diagnostic distribution. In contrast, Ji et al. reported increased consultation efficiency and reduced infections in rural populations following the adoption of telemedicine. For the outcome of feasibility, patient outcomes were measured through intervention assessments collected during and after implementation in both studies. These assessments gathered similar information from patients, including their daily symptoms, well-being, and general health. The implementation outcome of sustainability was uniquely evaluated by Pehlivan et al., based on the guiding framework, without requiring harmonization or standardized measures.

The Mixed Methods Appraisal Tool (MMAT) was employed to critically appraise these three articles, encompassing screening questions, qualitative assessments, and quantitative evaluations (Table 2). Following independent reviews and the completion of the appraisal process, each article achieved a score of 9 out of 12, signifying that 75% of the quality criteria were met.

Accorsi et al. yielded significant findings through their telemedicine intervention during the COVID-19 pandemic in Brazil, albeit with a relatively small sample size, which may limit the generalizability of their findings [18]. Similarly, Pehlivan et al.'s study in Turkey also featured a limited sample size [19]. In contrast, Ji et al.'s article boasted a more robust sample size in China, although it lacked a comprehensive description of costs [21].

Within each study, variation was present in the categorization of service outcomes as primary or secondary analysis. Within Ji et al. and Pehlivan et al., the service outcomes of efficacy, safety, equity, and patient-centricity were all evaluated for their impact following the implementation of telemedicine within the primary analysis of the study. These outcomes were interwoven throughout the result analysis, with some strengthening the effectiveness of others. In contrast, Accorsi et al. measured service outcomes as part of a secondary analysis, however, it was discussed that the silo effect of measuring the implementation outcome and

service outcome separately reduces the value in the prognosis of COVID-19 infection, as these outcomes tend to be mutually supportive.

## Discussion

The findings from all three articles, albeit somewhat limited, consistently underscored the feasibility and adoptability of telemedicine interventions, with limited attention to implementation outcomes such as cost and sustainability. Notably, these studies were conducted in middle-income countries, revealing an absence of telemedicine adoption in low-income countries as defined by the World Bank [7]. The common thread across all these articles was a focus on service delivery through telemedicine.

Among the three articles, it became evident that the effectiveness of telemedicine interventions was on par with face-to-face consultations. Particularly, rehabilitation delivered via telemedicine emerged as a cost-effective treatment approach compared to traditional in-person programs [19]. As a high-quality intervention for varied approaches, telemedicine expands access to treatment and facilitates the diagnosis of COVID-19 patients [18], all while minimizing risk for both patients and healthcare providers [21]. Furthermore, telemedicine stands out as a cost-effective approach, as it does not necessitate additional specialized equipment purchases by or for patients [19,21]. The lower expense and resource requirements for this versatile intervention make it suitable for use in some rural and underserved areas effectively reducing barriers to healthcare access [26].

However, it is worth noting a potential drawback: telemedicine may inadvertently create barriers to accessing certain diagnostics and medications for patients [27]. Across LICs, where the setting is primarily rural, the quality of the information and communications technologies (ICT) infrastructure might not adequately support remote monitoring and intervention [28]. In situations where immediate attention or diagnostic evaluation based on symptoms is imperative, prioritizing an in-person visit may be more judicious, as it can streamline the process by eliminating the need for two separate appointments—one via telemedicine and another in-person.

We are hopeful that further research and increased publications documenting the progress of telemedicine implementation efforts within low-income countries will emerge post-COVID. Understanding how infrastructure influences the feasibility, adoption, cost-effectiveness, and sustainability of telemedicine interventions is paramount. We advocate for efforts aimed at bolstering technological capabilities, increasing centralized community locations where telemedicine can be accessed, and developing secure platforms and mobile applications for patient use.

While many countries have expanded laws to permit the adoption of telemedicine in healthcare settings by issuing guidance at the national level [24,28] we urge for increased attention to existing regulatory frameworks and national guidance for utilizing secure digital health technologies in low-resource and remote settings. From a data privacy and security perspective, it is essential to comprehend how patient records are maintained and what resources are required to digitize these archives in a central repository accessible to necessary stakeholders. Ultimately, the potential benefits of present and future implementation of telemedicine significantly outweigh the barriers that need to be addressed [29].

### Limitations

Several limitations have impacted our assessment of the literature. One notable limitation in this systematic review is the scarcity of articles specifically focused on telemedicine and its implementation outcomes concerning COVID-19 in low-income countries. It is worth noting

that all the studies we extracted were centered on either upper- or lower-middle-income countries, implying a potential research gap concerning low-income countries in particular. Additionally, the relatively small number of articles retrieved suggests that limited research has been conducted in this field, possibly due to the recent emergence of COVID-19.

Turning our attention to the individual articles, Accorsi et al. acknowledged a study limitation related to its highly controlled environment, raising questions about the generalizability of the study results to the broader population [18]. Similar concerns regarding generalizability were also voiced in Pehlivan et al.'s study [19]. Furthermore, this study noted that many individuals faced challenges in utilizing technology and internet services, which could pose implementation issues for the telerehabilitation intervention among patients. In contrast, Ji et al.'s study stood out as the only one with a large sample size (n = 1500), making its population more likely to be generalizable [21]. However, it is important to emphasize that despite these limitations, the results across all three studies consistently highlighted the positive impact of telemedicine on both service and health outcomes among patients.

## Conclusion

Telemedicine emerges as a highly effective intervention for addressing the challenges posed by COVID-19 in LMICs. However, several critical factors warrant consideration when implementing this intervention. Many LMICs grapple with resource limitations, highlighting the need for adequate internet and technology infrastructure to facilitate telemedicine. It is crucial to recognize that telemedicine may not be a feasible strategy in some resource-constrained environments. In instances where telemedicine interventions are viable, they hold the potential to significantly mitigate the transmission of COVID-19 and empower patients to better manage the disease. Through this systematic review, it became evident that both patients and caregivers place substantial value on the benefits, safety, quality of care, and engaged care when seeking treatments that promote patient health. Telemedicine is one option to meet patient needs in a timely manner. Empirical evidence from randomized controlled trials underscores the effectiveness of telemedicine when deployed to assist individuals impacted by COVID-19.

Assessing technological capacity within communities should be a priority when implementing telemedicine interventions. In cases where patients lack internet access at home, thoughtful considerations should be given to supplying communities with the necessary computing resources and internet connectivity for teleconsultations and doctor-patient meetings. Local hospital administration can orchestrate such planning and implementation in collaboration with regional and national lawmakers, with due attention to the legal aspects of privacy guidelines and information dissemination between healthcare providers and patients.

Future research endeavors should be conducted in lower-income countries, encompassing larger sample sizes and a heightened focus on implementation research outcomes. The insights gleaned from these articles underscore the pressing need for additional research to comprehensively analyze telemedicine as an intervention for the management of COVID-19 in LMIC settings lacking robust technological resources.

## Supporting information

**S1 Appendix.  Systematic review search strategy.**
(DOCX)

**S1 Checklist.  Preferred Reporting Items for Systematic Reviews and Meta-Analyses (PRISMA) 2020 checklist.**
(PDF)

## Author contributions

**Conceptualization:** Nonye Marita Okafor, Imani Thompson, Vandana Venkat, Courtney Robinson.

**Data curation:** Nonye Marita Okafor, Imani Thompson, Vandana Venkat, Courtney Robinson, Hope Lappen, Dorice Vieira.

**Formal analysis:** Nonye Marita Okafor, Imani Thompson, Vandana Venkat, Courtney Robinson.

**Investigation:** Nonye Marita Okafor, Imani Thompson, Vandana Venkat, Courtney Robinson.

**Methodology:** Nonye Marita Okafor, Imani Thompson, Vandana Venkat, Courtney Robinson, Aishwarya Rao, Hope Lappen, Dorice Vieira, Emmanuel Peprah.

**Project administration:** Nonye Marita Okafor.

**Resources:** Nonye Marita Okafor, Imani Thompson, Vandana Venkat, Courtney Robinson, Aishwarya Rao, Emmanuel Peprah.

**Software:** Nonye Marita Okafor, Imani Thompson, Vandana Venkat, Courtney Robinson.

**Supervision:** Aishwarya Rao, John Pateña, Emmanuel Peprah.

**Validation:** Nonye Marita Okafor, Imani Thompson, Vandana Venkat, Courtney Robinson, Aishwarya Rao, Sumedha Kulkarni, Leah Frerichs, Khady Ndiaye, Deborah Adenikinju, Chukwuemeka Iloegbu, John Pateña, Hope Lappen, Dorice Vieira, Joyce Gyamfi, Emmanuel Peprah.

**Visualization:** Nonye Marita Okafor, Imani Thompson, Vandana Venkat, Courtney Robinson.

**Writing – original draft:** Nonye Marita Okafor, Imani Thompson, Vandana Venkat, Courtney Robinson.

**Writing – review & editing:** Nonye Marita Okafor, Imani Thompson, Vandana Venkat, Courtney Robinson, Aishwarya Rao, Sumedha Kulkarni, Leah Frerichs, Khady Ndiaye, Deborah Adenikinju, Chukwuemeka Iloegbu, John Pateña, Joyce Gyamfi, Emmanuel Peprah.

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
