## [Decision Letter · Decision Letter 0]

19 Apr 2024

PDIG-D-23-00476

Evaluating feasibility, adoption, cost-effectiveness, and sustainability of telemedicine interventions in managing COVID-19 within low-and-middle-income countries (LMICs): A systematic review.

PLOS Digital Health

Dear Dr. Peprah,

Thank you for submitting your manuscript to PLOS Digital Health. After careful consideration, we feel that it has merit but does not fully meet PLOS Digital Health's publication criteria as it currently stands. Therefore, we invite you to submit a revised version of the manuscript that addresses the points raised during the review process.

Please submit your revised manuscript within 60 days Jun 18 2024 11:59PM. If you will need more time than this to complete your revisions, please reply to this message or contact the journal office at digitalhealth@plos.org. Please include the following items when submitting your revised manuscript:

We look forward to receiving your revised manuscript.

Kind regards,

Erika Ong, M.D.

Guest Editor

PLOS Digital Health

Journal Requirements:

1. We ask that a manuscript source file is provided at Revision. Please upload your manuscript file as a .doc, .docx, .rtf or .tex.

Additional Editor Comments (if provided):

Dear Authors,

Thank you for the opportunity to review this interesting paper. Please address the comments and concerns of the reviewers to strengthen and support this manuscript's claims. I especially agree that the lack of representation from low income countries should be further discussed.

Reviewers' comments:

Reviewer's Responses to Questions

**Comments to the Author**

1. Does this manuscript meet PLOS Digital Health’s publication criteria ? Is the manuscript technically sound, and do the data support the conclusions? The manuscript must describe methodologically and ethically rigorous research with conclusions that are appropriately drawn based on the data presented.

Reviewer #1: Yes

Reviewer #2: Partly

2. Has the statistical analysis been performed appropriately and rigorously?

Reviewer #1: Yes

Reviewer #2: N/A

3. Have the authors made all data underlying the findings in their manuscript fully available (please refer to the Data Availability Statement at the start of the manuscript PDF file)?

Reviewer #1: Yes

Reviewer #2: Yes

4. Is the manuscript presented in an intelligible fashion and written in standard English?

PLOS Digital Health does not copyedit accepted manuscripts, so the language in submitted articles must be clear, correct, and unambiguous. Any typographical or grammatical errors should be corrected at revision, so please note any specific errors here.

Reviewer #1: Yes

Reviewer #2: Yes

5. Review Comments to the Author

Please use the space provided to explain your answers to the questions above. You may also include additional comments for the author, including concerns about dual publication, research ethics, or publication ethics. (Please upload your review as an attachment if it exceeds 20,000 characters)

Reviewer #1: I thank the Editors for the opportunity to review this work, which describes a meta-analysis to explore telehealth in LMICs during the Covid pandemic. The paper is well written. However, I have a few comments and concerns for consideration. 

1. The authors in the introduction describe telehealth as “seamless” or “critical.” I do not think there is enough evidence to support such strong language. Please temper the introduction considerably. Telehealth is by no means a panacea. 

2. In the methods, the authors write that “Subsequently, in (November 2023), we refined our inquiry by delving into public health and health science databases, enabling us to gain valuable insights into our target population, intervention, and outcome variables.” What prompted this change over a year later?

3. The authors make a good point that the three studies included – from Brazil, Turkey, and China – are from MICs. The authors should dedicate perhaps a paragraph or two to studies looking at telemedicine LMICs. Perhaps they can describe some of the papers that were selected out – might it be that the low representation of studies from LMICs is due to the lower feasibility of conducting telemedicine studies that would have been included in this meta-analysis given LMIC resource constraints? Taken a step further, this paper seeks to explore telemedicine in LMICs but seems to ‘select out’ papers from LMIC settings, in a way, shooting itself in the foot. I do not think the authors need to revise their methods at all. They just need to perhaps highlight this in the limitations if they agree with the above.

4. The authors could include in the discussion 1) a paragraph on next steps – what are actionable findings from their results? And 2) a section on telehealth in LIC and LMIC settings. 

5. It is interesting to note that all the authors and affiliations are in HICs. I wonder if this too is a limitation that should be addressed, especially if there is a sincere effort to “decolonize” global health research in the literature.

Reviewer #2: Suggest that the authors reconsider solely using the "taxonomy of implementation outcomes defined by Proctor et al. -- for feasibility, adoption, cost effectiveness and sustainability (of telemedicine) in this paper. A review of these concepts as these are used in the eHealth / digital health space must be evaluated, adopted for the study... 

Specifically: the authors wanted to investigate the adoption and sustainability of telemedicine --- the authors did not establish these 2 attributes of telemedicine -- and thus a better understanding of these concepts can enhance the authors' discourse

6. PLOS authors have the option to publish the peer review history of their article (what does this mean? ). If published, this will include your full peer review and any attached files.

**Do you want your identity to be public for this peer review?** For information about this choice, including consent withdrawal, please see our Privacy Policy .

Reviewer #1: No

Reviewer #2: Yes: Portia Grace F. Marcelo, MD MPH

Professor, University of the Philippines Manila

---

## [Decision Letter · Decision Letter 1]

7 Nov 2024

PDIG-D-23-00476R1

Evaluating feasibility, adoption, cost-effectiveness, and sustainability of telemedicine interventions in managing COVID-19 within low-and-middle-income countries (LMICs): A systematic review.

PLOS Digital Health

Dear Dr. Okafor,

Thank you for submitting your manuscript to PLOS Digital Health. After careful consideration, we feel that it has merit but does not fully meet PLOS Digital Health's publication criteria as it currently stands. Therefore, we invite you to submit a revised version of the manuscript that addresses the points raised during the review process.

Please submit your revised manuscript within 30 days Dec 07 2024 11:59PM. If you will need more time than this to complete your revisions, please reply to this message or contact the journal office at digitalhealth@plos.org. Please include the following items when submitting your revised manuscript:

We look forward to receiving your revised manuscript.

Kind regards,

Erika Ong, MD

Academic Editor

PLOS Digital Health

Journal Requirements:

1. As required by our policy on Data Availability, please ensure your manuscript or supplementary information includes the following: 

Additional Editor Comments (if provided):

Dear Authors,

Thank you for the comprehensive revisions you have made, and thank you very much for your patience as we sought another reviewer. Please see reviewer 3's very valuable comments. Your manuscript as it is is strong already, presents very relevant findings, and is almost ready for publication, but to realize its full potential I encourage you to consider the suggestions reviewer 3 laid out, namely, outlining aims/hypotheses more clearly in the beginning, elaborating more on how the implementation outcomes were measured, mentioning the service outcomes in the discussion as well since they were included it in the revision, and any potential interrelationships between implementation and service outcomes mentioned in the papers included. As this will likely be the last revision, please feel free to make final stylistic changes as you deem appropriate.

Thank you for submitting your manuscript to PLOS Digital Health, and we look forward to your revisions.

Kind Regards,

Erika Ong, MD

Academic Editor 

PLOS Digital Health

Reviewers' comments:

Reviewer's Responses to Questions

**Comments to the Author**

1. If the authors have adequately addressed your comments raised in a previous round of review and you feel that this manuscript is now acceptable for publication, you may indicate that here to bypass the “Comments to the Author” section, enter your conflict of interest statement in the “Confidential to Editor” section, and submit your "Accept" recommendation.

Reviewer #1: All comments have been addressed

Reviewer #3: (No Response)

2. Does this manuscript meet PLOS Digital Health’s publication criteria ? Is the manuscript technically sound, and do the data support the conclusions? The manuscript must describe methodologically and ethically rigorous research with conclusions that are appropriately drawn based on the data presented.

Reviewer #1: Yes

Reviewer #3: (No Response)

3. Has the statistical analysis been performed appropriately and rigorously?

Reviewer #1: Yes

Reviewer #3: (No Response)

4. Have the authors made all data underlying the findings in their manuscript fully available (please refer to the Data Availability Statement at the start of the manuscript PDF file)?

Reviewer #1: Yes

Reviewer #3: (No Response)

5. Is the manuscript presented in an intelligible fashion and written in standard English?

PLOS Digital Health does not copyedit accepted manuscripts, so the language in submitted articles must be clear, correct, and unambiguous. Any typographical or grammatical errors should be corrected at revision, so please note any specific errors here.

Reviewer #1: Yes

Reviewer #3: (No Response)

6. Review Comments to the Author

Please use the space provided to explain your answers to the questions above. You may also include additional comments for the author, including concerns about dual publication, research ethics, or publication ethics. (Please upload your review as an attachment if it exceeds 20,000 characters)

Reviewer #1: Thank you for your hard work addressing the suggested revisions. The paper is in great shape, with limitations appropriately acknowledged.

Reviewer #3: Evaluating feasibility, adoption, cost-effectiveness, and sustainability of telemedicine interventions in managing COVID-19 within low- and middle-income countries (LMICs): A Systematic Review

Given the importance of implementing evidence-based interventions for pandemics such as COVID-19 and their urgency in low-resource settings, the paper examines the feasibility, adoption, cost-effectiveness, and sustainability of telemedicine interventions. While a growing literature has addressed both the use of telemedicine and the urgency of implementing proven interventions for pandemic, this paper is one of the few that examines the implementation outcomes of such interventions. Attaining implementation outcomes is important for two reasons: (1) we have little data about the telemedicine’s feasibility, adoption, cost-effectiveness, and sustainability; and (2) ensuring that those outcomes are attained is critical for subsequent tests of telemedicine’s clinical effectiveness (one cannot conclude that telemedicine is effective without ensuring that it was implemented well). While claiming the importance of attaining implementation outcomes is not novel, direct examination of their implementation via the lens of implementation outcomes is under-studied. 

Relevant literature is succinctly and adequately reported. 

The manuscript would be strengthened by a list of the (clearly stated or enumerated) research aims, questions, or hypotheses that the systematic review aims to inform. 

Methods for the systematic review are described clearly, including the search strategy, inclusion and exclusion criteria, and data extraction methods. Moreover, the authors conducted a bias risk assessment using the Mixed methods Appraisal Tool. 

The scoring method and interpretation of colors for cells in Table 2 need better explanation. 

Likewise, this reviewer requests more detail about how each of the implementation outcomes was measured in each study. Were standardized measures employed? Who reported on implementation outcome attainment? What data sources? How harmonized was measurement of a given outcome across studies? 

Table 3 reports the service outcomes addressed in each study, but the literature review and manuscript framing focus primarily on implementation outcomes. Including the service outcomes increases the manuscript’s value, but more discussion is needed. Why were these outcomes included in data reporting? Did any of the three papers address inter-relationships among implementation outcomes, or any analysis of association between implementation and service outcomes? 

In the same vein, the authors report available effectiveness data. This reviewer requests that the manuscript frame the aims/ questions/ hypotheses driving the systematic review before the methods section and include service outcomes and effectiveness outcomes in those aims. 

• the manuscript warrant such treatment?

• If the paper is considered unsuitable for publication in its present form, does the study itself show sufficient potential that the authors should be encouraged to resubmit a revised version?

This manuscript’s methods and findings, albeit limited as the authors acknowledge, are important additions to the scant evidence about the topic. However more detail as noted above will strengthen the paper’s contributions.

7. PLOS authors have the option to publish the peer review history of their article (what does this mean? ). If published, this will include your full peer review and any attached files.

**Do you want your identity to be public for this peer review?** For information about this choice, including consent withdrawal, please see our Privacy Policy . 

Reviewer #1: No

Reviewer #3: None

---

## [Editor Report · Decision Letter 2]

31 Jan 2025

Evaluating feasibility, adoption, cost-effectiveness, and sustainability of telemedicine interventions in managing COVID-19 within low-and-middle-income countries (LMICs): A systematic review.

PDIG-D-23-00476R2

Dear Dr Peprah,

We are pleased to inform you that your manuscript 'Evaluating feasibility, adoption, cost-effectiveness, and sustainability of telemedicine interventions in managing COVID-19 within low-and-middle-income countries (LMICs): A systematic review.' has been provisionally accepted for publication in PLOS Digital Health.

Best regards,

Erika Ong

Academic Editor

PLOS Digital Health

**Additional Editor Comments (if provided):**

Please correct repeated sentence in Page 6 Lines 144-148.